# Forage Radish Cover Crops Improve Soil Quality and Fruit Yield of *Lycium barbarum* L. in an Arid Area of Northwest China

Fang Wang [1,2,†], Wenhui Li [1,2,†], Haonan Chen [1,2], Ray R. Weil [3], Lizhen Zhu [4] and Xiongxiong Nan [4,*]

[1] College of Geographical Sciences and Planning, Ningxia University, Yinchuan 750021, China; fangwang0820@nxu.edu.cn (F.W.); liwenhuiyolo@163.com (W.L.); chn1504924790@163.com (H.C.)

[2] China-Arab Joint International Research Laboratory for Featured Resources and Environmental Governance in Arid Regions, Yinchuan 750021, China

[3] Department of Environmental Science and Technology, University of Maryland, College Park, MD 20742, USA; rweil@umd.edu

[4] State Key Laboratory of Efficient Production of Forest Resources, Yinchuan 750002, China; lizhenzhu916@163.com

* Correspondence: nanxiong0820@163.com; Tel./Fax: +86-951-5667118

† These authors contributed equally to this work.

**Abstract:** Intercropping orchards with cover crops is an important practice for achieving sustainable soil management. However, little research has addressed the development of a soil quality index (SQI) to evaluate cover crop effects on orchard soil quality. The aim of this study was to ascertain whether cover cropping improves soil quality and fruit yield of Goji (*Lycium barbarum* L.) while reducing or replacing organic fertilizer application. The main treatments were the traditional management of *L. barbarum* as a monocrop (M) and intercropping Goji with radish (*Raphanus sativus* L.) as an annual cover crop (I). Within the main treatments, different levels of organic fertilizer were applied at $0\ kg\cdot plant^{-1}$ ($M_0$), $2\ kg\cdot plant^{-1}$ ($M_1$), and $4\ kg\cdot plant^{-1}$ ($M_2$). After six years of planting, we analyzed the changes in soil quality caused by cover cropping with different organic fertilizer levels based on the SQI method. Goji yields were used for validation of the SQI derived from a minimum data set of soil quality indicators. In contrast with traditional monocropping, cover cropping increased soil total nitrogen, available nitrogen, and available phosphorus contents (by 78.60%, 30.30%, and 138.08%, respectively). There were also increased microbial biomass carbon and nitrogen contents (by 79.01% and 184.01%, respectively), enhanced urease and sucrase activities (by 41.02% and 56.81%, respectively), and reduced bulk density (by 1.92%) in the soil as a result of cover cropping. Compared with $IM_0$ treatment, soil microbial biomass carbon and nitrogen contents considerably increased under $IM_1$ treatment, whereas soil available nitrogen and potassium contents as well as electrical conductivity increased under $IM_2$ treatment. The SQI, which varied among treatments in the order $IM_1 > IM_2 > MM_2 > MM_1 > IM_0 > MM_0$, was positively correlated with Goji yield. From the soil quality and Goji yield perspective, cover cropping with a medium level of organic fertilizer is the optimal soil management practice for the *L. barbarum* planting system in arid areas of Ningxia, Northwest China.

**Keywords:** crop yield; minimum data set; organic manure; *Raphanus sativus* L.; radish cover crop; soil quality index; structural equation model

## 1. Introduction

Goji (*Lycium barbarum* L.), a perennial shrub in the Solanaceae family, is favored by Chinese consumers as a functional resource of the 'homology of medicine and food' [1]. To meet the growing market demand, the main production areas of *L. barbarum*—which is a characteristic cash crop in Northwest China—have expanded from Ningxia to Gansu, Qinghai, and Xinjiang. By the end of 2021, the total planting area and output of *L. barbarum*

in China reached $2 \times 10^5$ hm$^2$ and $4.41 \times 10^5$ t, respectively. As an advantageous and featured industry, *L. barbarum* production plays an essential role in driving regional economic growth, promoting rural revitalization, and increasing farmers' income.

However, long-term planting of *L. barbarum* is associated with soil deterioration and consequently the decline in fruit yield and quality, resulting in low economic and ecological benefits. This is mainly due to the implementation of extensive management practices, such as traditional monocropping, excessive chemical fertilizer application, insufficient organic amendment, and flood irrigation [2].

Cover cropping in orchards is an advanced soil management practice and a sustainable development model for improving orchard quality and efficiency [3]. It conveys benefits to fruit yield and quality by increasing water infiltration, improving soil fertility [4,5], and promoting soil nitrogen cycling and carbon storage [6]. Hence, incorporating *L. barbarum* orchards with high-quality cover crops would facilitate sustainable production. Forage radish (*Raphanus sativus* L. *var. longipinnatus*) is an annual cover crop that exhibits a complementary effect with *L. barbarum* in terms of water and fertilizer demand. Recently, a sustainable planting system of *L. barbarum* intercropped with *R. sativus* has been established from a new perspective [1], but its effects on the sustainability of soil quality and fruit yield still need to be ascertained.

Soil quality reflects the level of soil production capacity [7]. Soil quality assessment considers the impacts of soil management practices, land use changes, and other human activities. The assessment results assist in understanding the status quo and dynamics of soil quality in a timely manner, which is essential for achieving sustainable management of land resources [8,9]. Myriad studies have characterized the correlation between specific soil properties (or environmental factors) and crop yield under fertilizer application, making it difficult to depict a full picture of soil quality. This necessitates the development of a more objective and holistic method for accurate soil quality assessment. For instance, a soil quality index (SQI) can be established with representative soil quality indicators selected by principal component analysis (PCA). SQI-based soil quality assessment can accurately identify the impacts of soil management practices, and as such, it can provide guidance for sustainable production [10].

Chen et al. (2021) [11] selected a set of soil physical, chemical, and biological indicators using PCA to assess soil quality in a 14-year field experiment, and they found that straw mulching with 240 kg·hm$^2$ of nitrogen fertilizer was the optimum fertilization regime for farmland soil. Additionally, Jin et al. (2021) [12] proposed a practical, time-saving, and cost-effective method for quantitatively assessing the susceptibility of purple soils to water erosion in hilly areas worldwide, and this method was based on soil quality assessment of the cultivated horizon with varied erosion levels in sloping farmland in China. Furthermore, Liu et al. (2005) [13] established a minimum data set (MDS) through PCA to assess the soil quality of forestland, including natural secondary forests and four types of pure forests (i.e., tea-oil tree, yellow peach, arbutus, Chinese fir) in subtropical regions. While existing studies have demonstrated the effects of fertilization regime and soil type in farmland and natural vegetation restoration in forestland, less is known about how organic fertilizer and cover cropping affects soil quality in orchards in arid areas such as Ningxia.

Here, *L. barbarum* was intercropped with *R. sativus* in an arid area of Ningxia, China. The traditional planting system of *L. barbarum* (monocropping) was compared with the sustainable planting system of *L. barbarum*/*R. sativus* (intercropping) in terms of soil physical, chemical, and biological properties, as well as crop yield. The objectives of the present study were to: (1) evaluate the changes in soil quality caused by cover cropping with different organic fertilizer levels in *L. barbarum* orchards based on the SQI method and (2) ascertain whether cover cropping improves soil quality and fruit yield while reducing or replacing the application of organic fertilizer. The findings of this study will inform the identification of optimum practices for sustainable soil management in *L. barbarum* fields in arid areas.

## 2. Materials and Methods

### 2.1. Field Site Description

The field experiment was carried out in 2016 at the demonstration base (38°24′ N, 106°10′ E) of Goji Engineering Technology Research Center, State Forestry Administration of China (Yinchuan, Ningxia, China). The experimental site, which has an average altitude of approximately 1110 m, is located in a warm temperate continental monsoon climate zone with dry and windy winters and springs. Its average annual temperature is 8.5 °C, and the frost-free period lasts 160–170 days. The average annual precipitation and evaporation are 200 and 1883 mm, respectively. The major soil type is aeolian sandy soil. The initial soil before the start of the experiment had a bulk density of 1.45 g·cm$^{-3}$, with 67.63% sand, 15.44% silt, and 16.93% clay. The soil was alkaline with a pH of 9.31 and contained 8.11 g·kg$^{-1}$ organic carbon (SOC), 0.72 g·kg$^{-1}$ total nitrogen (TN), 18.21 mg·kg$^{-1}$ available nitrogen (AN), 16.65 mg·kg$^{-1}$ available phosphorus (AP), and 47.88 mg·kg$^{-1}$ available potassium (AK).

### 2.2. Experimental Design and Yield Estimation

The experiment used a randomized complete block design arranged in split-plots. The main plots included two cropping systems, and the subplots included three organic fertilizer levels. In total, six treatments were used, i.e., *L. barbarum* monocropping with no organic fertilizer (MM$_0$), medium-level organic fertilizer (2 kg·plant$^{-1}$; MM$_1$), and high-level organic fertilizer (4 kg·plant$^{-1}$; MM$_2$), and *L. barbarum/R. sativus* intercropping with no organic fertilizer (IM$_0$), medium-level organic fertilizer (2 kg·plant$^{-1}$; IM$_1$), and high-level organic fertilizer (4 kg·plant$^{-1}$; IM$_2$). There were three replicate blocks, each with six 20 m × 6 m plots (one treatment per plot).

Chemical fertilizers—urea, diammonium phosphate, and potassium sulfate—were top-dressed at the roots of *L. barbarum* by hole application from late April to late June and from the end of June to late July. Briefly, a shovel was used to dig three to four holes under the edge of the crown of *L. barbarum*. After the fertilizers were applied, the hole was backfilled with soil. Each plant received 120 g of N, 60 g of P$_2$O$_5$, and 75 g of K$_2$O from one topdressing, with a total of two topdressings in the whole growth period. The levels of chemical fertilizer application for each topdressing were 400 kg·hm$^{-2}$ N, 200 kg·hm$^{-2}$ P$_2$O$_5$, and 250 kg·hm$^{-2}$ K$_2$O. Organic fertilizer—decomposed sheep manure—was applied through a ring-shaped ditch (30 cm deep, 30 cm wide) at a distance of 30 cm from the trunk of *L. barbarum* during the leaf expansion period. The organic fertilizer contained 44.60% moisture, 12.44% organic carbon, 1.10% nitrogen, 0.32% P$_2$O$_5$, and 0.67% K$_2$O. The levels of organic fertilizer application were 0, 6660, and 13,320 kg·hm$^{-2}$.

In the traditional monocropping system, the row spacing of *L. barbarum* was 1 m × 3 m (Figure 1a), and no crops were planted between rows, with only normal weeding and management. In the intercropping system, *R. sativus* was interplanted along each row of *L. barbarum* at the beginning of autumn (early August)—the late growth (summer dormancy) period of *L. barbarum* (Figure 1b). The plant spacing of *R. sativus* was 20 cm, with a distance of 20 cm from *L. barbarum*. *R. sativus* seeds were sown to a depth of 1–2 cm at the density of 33,300–37,500 plants·hm$^{-2}$. All cover crops were retained in the field, frosted to death in late December, and decomposed in the following spring. The remaining field management practices in the intercropping system were the same as those implemented in the traditional monocropping system.

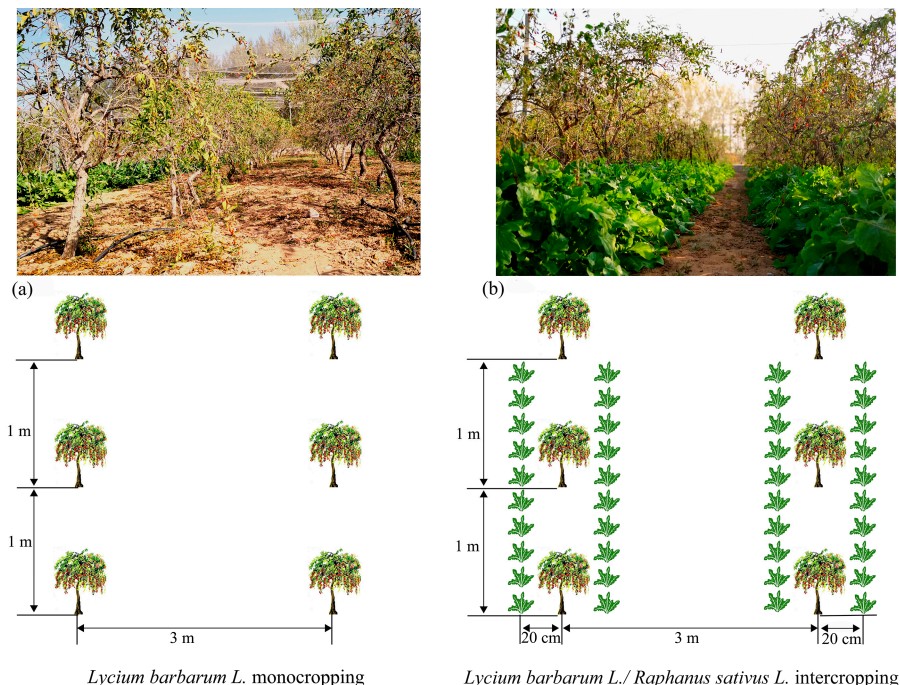

*Lycium barbarum L.* monocropping      *Lycium barbarum L./ Raphanus sativus L.* intercropping

**Figure 1.** Schematic diagram and photographs of the field experiment conducted on the demonstration base of Goji Engineering Technology Research Center, State Forestry Administration of China (Yinchuan, Ningxia, China). (**a**) Traditional *L. barbarum* monocropping and (**b**) *L. barbarum/Raphanus sativus* intercropping.

In 2021, *L. barbarum* fruits were collected when the first batch matured, and all fruits were continuously harvested to evaluate the annual yield. Five plants per plot were selected at random to determine their cumulative yield from late June to mid-September, and the yield per plant was used to calculate the yield per unit area.

### 2.3. Soil Sampling and Laboratory Analysis

Soil samples were collected after harvest of *L. barbarum* in early August 2021. In each plot, five representative sampling points were selected, and the samples (0–20 cm depth) were composited evenly. After removing the debris (e.g., gravel, plant roots), the samples were passed through a 2 mm sieve. One quarter of each fresh sample was used to analyze soil AN, enzyme activity, and microbial biomass carbon (MBC) and nitrogen (MBN) contents. The remaining soil samples were air-dried and used for the analysis of basic physicochemical properties.

Standard testing methods were used to determine soil properties [14]. Briefly, bulk density (BD) was measured by the cutting ring method. SOC and TN contents were determined by potassium dichromate oxidation–external heating and Kjeldahl methods, respectively. AN, AP, and AK contents were analyzed in 1 mol·L$^{-1}$ KCl extracts with a flow analyzer, 0.5 mol·L$^{-1}$ NaHCO$_3$ extracts with molybdenum-antimony colorimetry, and 1 mol·L$^{-1}$ NH$_4$OAc extracts by flame photometry, respectively. The pH and electrical conductivity (EC) were, respectively, measured using potentiometric and conductometric methods with a water–soil ratio of 2.5:1 (*v/w*). MBC and MBN contents were determined by chloroform fumigation. Urease, alkaline phosphatase, and sucrase activities were assayed by the indophenol colorimetry, sodium thiosulfate titration, and benzene disodium phosphate colorimetry, respectively [15].

*2.4. Soil Quality Assessment*

An SQI was established through the following three steps: (1) selecting MDS indicators that could best represent soil functions and processes, (2) scoring soil quality indicators based on a nonlinear scoring equation, and (3) integrating indicator scores into the SQI.

2.4.1. MDS Determination

First, soil quality indicators were selected from principal components with an eigenvalue $\geq 1$ and the absolute value of factor loading $\geq 0.5$. In this process, the maximum variance rotation was used to enhance the explanatory power of uncorrelated components. Moreover, if the loading of an indicator was >0.5 for two principal components, correlation analysis was conducted, and the respective indicator was merged into the group where it had a lower correlation with other indicators. Then, the norm value of soil quality indicators in each group was calculated (Equation (1)), and the indicators with the norm value in the range of 10% of the maximum value were selected. If multiple indicators were retained in a given group, Pearson correlation coefficient was used to determine whether each of the indicators should be retained. If the correlation was low (*rho* < 0.5), each indicator was included in the MDS. If the indicators were significantly correlated in the principal components (*rho* > 0.5), the indicator with the largest norm value was selected to enter the MDS [16]. Finally, the MDS indicators were normalized and scored using Equation (2) [17].

$$N_{ik} = \sqrt{\sum_{j=1}^{k} \left( u_{ik}^2 e_k \right)} \tag{1}$$

where $N_{ik}$ is the norm value of the i-th indicator in the first k principal components with an eigenvalue >1; $u_{ik}$ is the load of the i-th indicator in the k-th principal component; and $e_k$ is the eigenvalue of the k-th principal component.

Each soil quality indicator was converted to a dimensionless value using a nonlinear scoring equation (Equation (2)) [17].

$$S = \frac{a}{1 + (x/x_0)^b} \tag{2}$$

where S is the nonlinear score (S = 0–1); a is the maximum score (a = 1); x is the measured value of each indicator; $x_0$ is the mean value of each indicator; and b is the slope of the equation. From the soil productivity and sustainability perspective, an appropriate scoring algorithm was selected and explained. When the indicators were considered beneficial to soil quality, they were used in ascending (the higher the better) and descending (the lower the better) orders, assigned values of –2.5 and 2.5, respectively [17].

The transformed indicator scores were integrated into an SQI by the weighting method (Equation (3)).

$$SQI = \sum_{i=1}^{n} W_i \times S_i \tag{3}$$

where $SQI_i$ is the soil quality index; $S_i$ is the score of the i-th indicator; $W_i$ is the ratio of the communality of the i-th indicator to the total variance in PCA; and n is the number of soil quality indicators. Higher SQI values are associated with better soil functions and processes, reflecting the positive impacts of land-use change in terms of nutrient cycling, soil resistance, and resilience, as well as soil productivity and sustainability [17]. Based on SQI values, the soil quality in the study area was classified into five grades: low (0–0.2), relatively low (0.2–0.4), moderate (0.4–0.6), good (0.6–0.8), and excellent (0.8–1.0).

2.4.2. MDS Verification

SQI values were calculated based on the total data set (TDS) and MDS with soil quality indicator scores and their weights under different treatments. Then, the SQIs of the two data sets were subjected to linear regression analysis to evaluate the accuracy of the MDS-based SQI.

### 2.5. Statistical Analysis

*Statistical analyses* were performed using IBM SPSS software (version 24.0, IBM Corp., Armonk, NY, USA). Two-way analysis of variance (ANOVA) was used to evaluate the effects of cropping system, organic fertilizer level, and the interactions between them on soil properties and fruit yield. Comparisons between group means were performed at the 5% level using Duncan's multiple range test. PCA was used to calculate the weights of selected variables among various soil quality indicators. Radar charts of variable loadings and norm values were plotted in Origin Pro 2021 (OriginLab Corp., Northampton, MA, USA). Structural equation modeling (SEM) was implemented in IBM SPSS Amos (version 24) to determine the relative importance of selected variables and their relationships.

## 3. Results

### 3.1. Soil Physicochemical Properties

Two-way ANOVA results showed that different cropping systems significantly affected soil BD, TN, AN, and AP (Table 1). These four soil properties changed variably in the intercropping system compared with the monocropping system. TN, AN, and AP contents increased under cover cropping by 78.60%, 30.30%, and 138.08%, respectively ($p < 0.05$), which contradicted to the decrease in soil BD by 1.92%.

**Table 1.** Soil physicochemical properties in various cropping systems under different organic fertilizer levels.

| Treatment | BD (g·cm³) | SOC (g·kg⁻¹) | TN (g·kg⁻¹) | AN (mg·kg⁻¹) | AP (mg·kg⁻¹) | AK (mg·kg⁻¹) | pH | EC (μS·cm⁻¹) |
|---|---|---|---|---|---|---|---|---|
| $MM_0$ | 1.45 ± 0.01 a | 9.04 ± 2.52 a | 0.76 ± 0.24 b | 19.71 ± 7.87 b | 15.41 ± 9.74 b | 52.65 ± 25.71 b | 8.97 ± 0.28 a | 109.37 ± 12.81 c |
| $MM_1$ | 1.45 ± 0.01 a | 9.11 ± 2.79 a | 0.71 ± 0.25 b | 35.20 ± 5.74 ab | 17.05 ± 4.92 b | 38.89 ± 11.42 b | 8.85 ± 0.15 a | 133.33 ± 16.98 bc |
| $MM_2$ | 1.45 ± 0.02 a | 11.25 ± 3.38 a | 0.88 ± 0.44 ab | 35.2 ± 12.03 ab | 15.84 ± 7.12 b | 78.04 ± 35.74 ab | 8.93 ± 0.34 a | 155.6 ± 30.32 b |
| $IM_0$ | 1.42 ± 0.02 ab | 11.04 ± 0.80 a | 1.34 ± 0.12 ab | 27.21 ± 2.71 bc | 35.96 ± 10.51 ab | 51.00 ± 0.72 b | 9.01 ± 0.02 a | 116.80 ± 4.79 c |
| $IM_1$ | 1.40 ± 0.03 b | 13.21 ± 1.85 a | 1.51 ± 0.55 a | 40.69 ± 5.28 ab | 31.39 ± 15.55 ab | 74.59 ± 18.55 ab | 8.97 ± 0.17 a | 136.23 ± 22.61 bc |
| $IM_2$ | 1.45 ± 0.01 a | 11.39 ± 0.64 a | 1.33 ± 0.28 ab | 49.50 ± 8.37 a | 47.63 ± 15.77 a | 114.47 ± 31.43 a | 8.85 ± 0.13 a | 191.07 ± 10.57 a |
| A | * | ns | ** | * | ** | ns | ns | ns |
| B | ns | ns | ns | ** | ns | * | ns | *** |

BD: bulk density; SOC: soil organic carbon; TN: total nitrogen; AN: available nitrogen; AP: available phosphorus; AK: available potassium; EC: electrical conductivity. $MM_0$, $MM_1$, and $MM_2$ represent *L. barbarum* monocropping with zero, medium, and high levels of organic fertilizer, respectively; $IM_0$, $IM_1$, and $IM_2$ represent *L. barbarum/R. sativus* intercropping with zero, medium, and high levels of organic fertilizer, respectively. A: Cropping system; B: Organic fertilizer level (ns: not significant, * $p < 0.05$, ** $p < 0.01$, and *** $p < 0.001$). Data represent the means ($n = 3$). Values followed by the same lowercase letters in a row are not significantly different between treatments (Duncan's multiple range test: $p < 0.05$).

Organic fertilizer level exhibited a significant effect on soil AN, AK, and EC (Table 1). In the monocropping system, the AN and EC of $MM_1$ treatment increased by 78.59% and 21.91%, respectively, in contrast to the 26.13% decrease in AK content; however, these differences were not statistically significant when compared with $MM_0$ treatment. Additionally, the AN, AK, and EC of $MM_2$ treatment increased by 78.58%, 48.22%, and 42.27%, respectively, when compared with those of $MM_0$ treatment; among them, only the difference in EC reached the significant level ($p < 0.05$). In the intercropping system, AN, AK, and EC all increased noticeably with increasing level of organic fertilizer. Compared with $IM_0$ treatment, the AN, AK, and EC of $IM_1$ treatment increased by 49.53%, 46.26%, and 16.64%, respectively, albeit not statistically significant. More prominent increase occurred in the AN, AK, and EC of $IM_2$ treatment by 81.93%, 124.46%, and 63.58%, respectively ($p$-values $< 0.05$).

The indicator sensitivity was classified using the coefficient of variation (CV). AP and AK contents were moderately sensitive soil quality indicators (CV = 0.58 and 0.47, respectively). SOC, TN, AN, and EC were low-sensitivity indicators (CV = 0.1–0.4). BD and pH were both insensitive indicators not suitable for selection into the MDS (CV = 0.02 and 0.02, respectively).

### 3.2. Soil Biological Properties

The cropping system and organic fertilizer level, as well as their interactions, exhibited significant effects on soil MBC and MBN contents (Figure 2). Compared with the monocropping system, the MBC and MBN contents, together with urease and sucrase activities, increased by 79.01%, 184.01%, 41.02%, and 56.81%, respectively, in the intercropping system. Soil biological properties did not change significantly with an increasing level of organic fertilizer under monocropping. However, compared with $IM_0$ treatment under intercropping, soil MBC and MBN contents of $IM_1$ treatment increased by 212.98% and 260.06%, respectively ($p$-values < 0.05), and those of $IM_2$ treatment increased by 135.76% ($p$ < 0.05) and 58.36%, respectively. MBC and MBN contents, as well as sucrase activity, were moderately sensitive soil quality indicators (CV = 0.43–0.83). Urease and phosphatase activities were low-sensitivity indicators (CV = 0.29 and 0.40, respectively).

### 3.3. MDS Construction for Soil Quality Indicators

A total of 13 soil quality indicators were screened by PCA, and the principal components with eigenvalues ≥1 were selected. The first four principal components showed a strong explanatory power (cumulative variance explained: 82.98%; Figure 3a). All indicators were divided into three groups. The first group consisted of SOC, TN, MBC, and MBN contents, as well as urease activity; the second group comprised AN, AP, and AK contents, plus EC; the third group contained phosphatase and sucrase activities. In the first group, the norm values of various indicators ranked as follows: urease > MBN > SOC > TN > MBC, all of which were within the 10% range of the maximum value. Among them, urease activity was retained in the MDS because of its significant correlation with MBN, SOC, and TN contents. Similarly, AN content in the second group and phosphatase activity in the third group entered the MDS. As such, three MDS indicators—urease activity, AN content, and phosphatase activity—were selected.

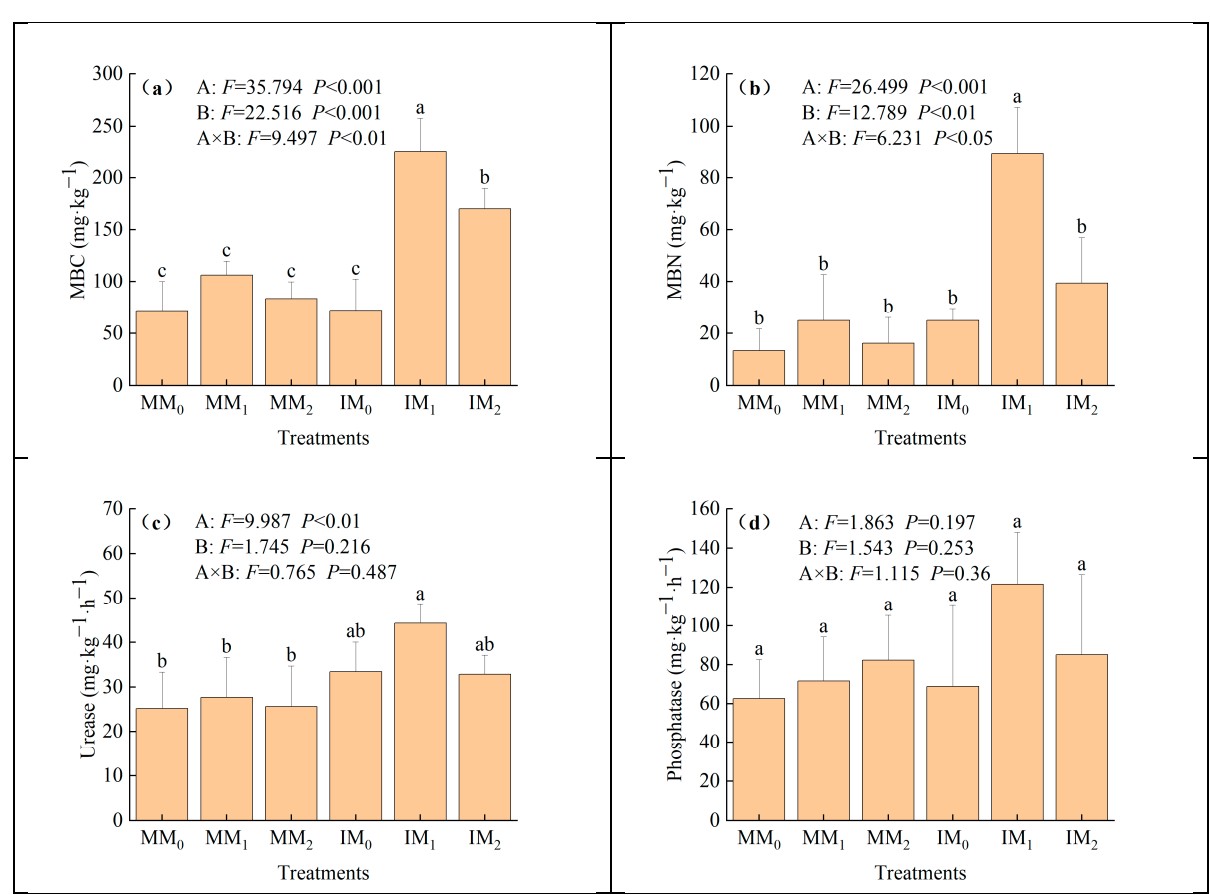

**Figure 2.** *Cont.*

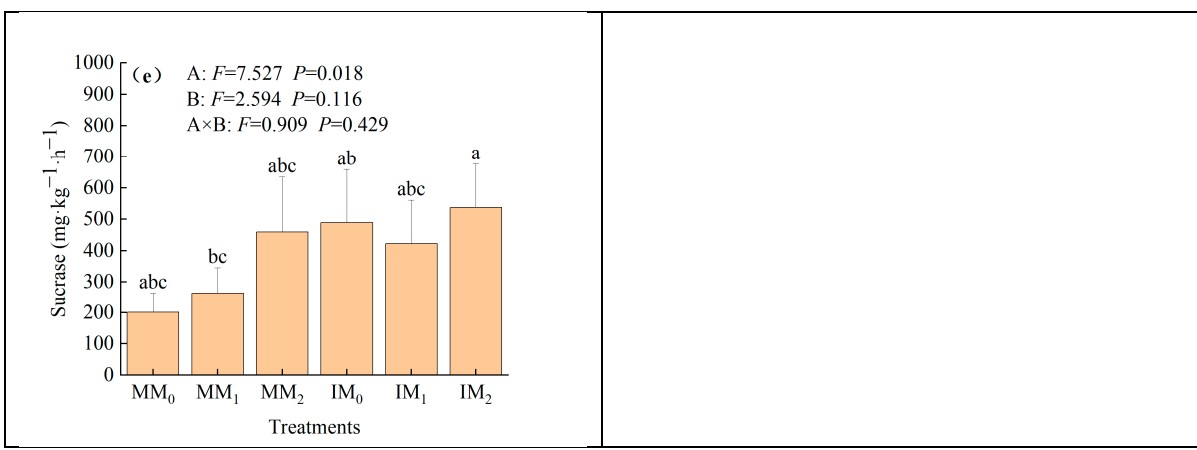

**Figure 2.** Soil biological properties in various cropping systems under different organic fertilizer levels. (**a**) Microbial biomass carbon (MBC) content; (**b**) microbial biomass nitrogen (MBN) content; (**c**) urease activity; (**d**) phosphatase activity; (**e**) urease activity. (MM$_0$, MM$_1$, and MM$_2$ represent *L. barbarum* monocropping with zero, medium, and high levels of organic fertilizer, respectively; IM$_0$, IM$_1$, and IM$_2$ represent *L. barbarum/R. sativus* intercropping with zero, medium, and high levels of organic fertilizer, respectively. A: cropping system; B: organic fertilizer level. Error bars represent standard error of the means (*n* = 3). The same lowercase letters above error bars indicate no significant difference between treatments (Duncan's multiple range test: *p* < 0.05)).

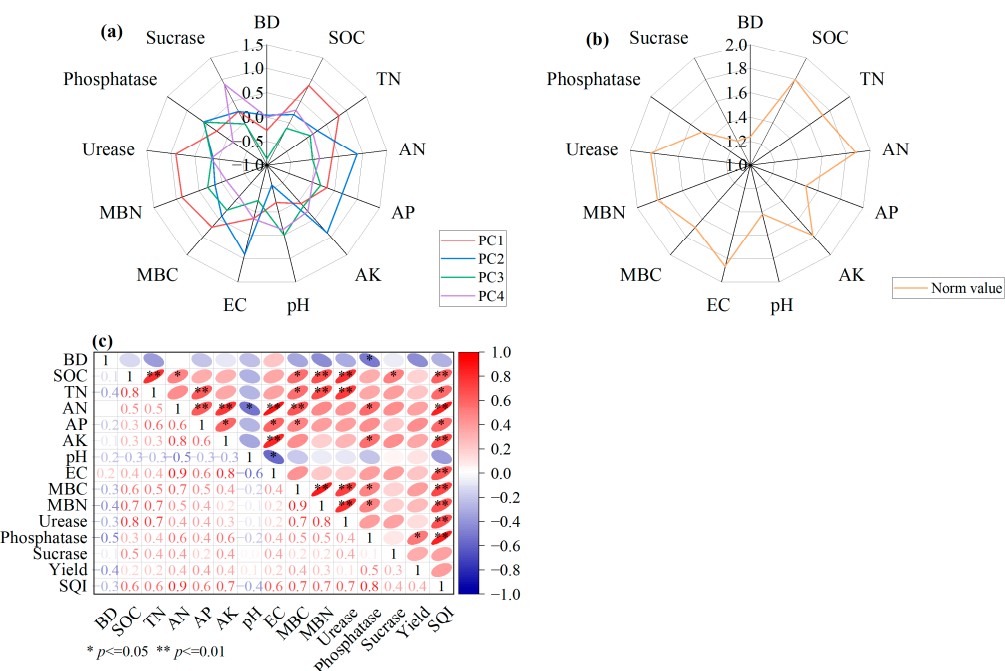

**Figure 3.** Minimum data set construction for soil quality indicators. (**a**) Radar chart of variable loading. (**b**) Radar chart of norm values. (**c**) Correlation heatmap of soil properties, fruit yield, and soil quality index (SQI). (BD: bulk density; SOC: soil organic carbon; TN: total nitrogen; AN: available nitrogen; AP: available phosphorus; AK: available potassium; EC: electrical conductivity; MBC, microbial biomass carbon; MBN, microbial biomass nitrogen.)

### *3.4. MDS-Based Soil Quality Assessment*

The communalities and weights of soil quality indicators in TDS and MDS were obtained *based on* PCA (Table 2). The indicator values were normalized to dimensionless values between 0–1 by a nonlinear scoring equation. pH and BD reflect soil alkalinity and

permeability, respectively, which conform to 'the lower the better' type function. Other indicators were major nutrient contents belonging to 'the higher the better'-type function. The weights of AN content and urease and phosphatase activities were 0.370, 0.276, and 0.354, respectively, indicating their positive effects on soil quality.

**Table 2.** Indicator weights of different data sets for soil quality assessment.

| Indicator | TDS | | MDS | |
|---|---|---|---|---|
| | Communality | Weight | Communality | Weight |
| BD | 0.824 | 0.076 | | |
| SOC | 0.878 | 0.081 | | |
| TN | 0.770 | 0.071 | | |
| AN | 0.911 | 0.084 | 0.716 | 0.370 |
| AP | 0.590 | 0.055 | | |
| AK | 0.876 | 0.081 | | |
| pH | 0.764 | 0.071 | | |
| EC | 0.946 | 0.088 | | |
| MBC | 0.770 | 0.071 | | |
| MBN | 0.895 | 0.083 | | |
| Urease | 0.864 | 0.080 | 0.535 | 0.276 |
| Phosphatase | 0.773 | 0.072 | 0.686 | 0.354 |
| Sucrase | 0.926 | 0.086 | | |

BD: bulk density; SOC: soil organic carbon; TN: total nitrogen; AN: available nitrogen; AP: available phosphorus; AK: available potassium; EC: electrical conductivity; MBC: microbial biomass carbon; MBN: microbial biomass nitrogen. TDS: total data set; MDS: minimum data set.

The non-linear scoring function was used to obtain the score of MDS indicators (Figure 4). Among them, AN (Figure 4a) had the highest score (0.26) in $IM_2$ treatment, with the lowest score (0.08) in $MM_0$ treatment. Compared with $MM_0$, urease scores increased by 51.71%, 99.90% ($p < 0.05$), and 51.71% in $IM_0$, $IM_1$, and $IM_2$ treatments, respectively (Figure 4b). Phosphatase scores had minimal differences among various treatments (Figure 4c).

The SQI based on TDS and MDS can be expressed as follows:

$$\text{SQI-TDS} = \text{BD} \times 0.076 + \text{SOC} \times 0.081 + \text{TN} \times 0.081 + \text{AN} \times 0.084 + \text{AP} \times 0.055 + \text{AK} \times 0.081 + \text{pH} \times 0.071 + \text{EC} \times 0.088 + \text{MBC} \times 0.071 + \text{MBN} \times 0.083 + \text{urease} \times 0.080 + \text{phosphatase} \times 0.072 + \text{sucrase} \times 0.086;$$

$$\text{SQI-MDS} = \text{AN} \times 0.370 + \text{urease} \times 0.276 + \text{phosphatase} \times 0.354.$$

Based on both of the TDS and MDS, the SQI values of the intercropping system were significantly higher than those of the monocropping system by 51.16% and 37.85%, respectively ($p$-values $< 0.05$, Figure 5). This means that cover cropping was superior to traditional monocropping in terms of soil quality improvement. A high correlation emerged between the SQI values based on TDS and MDS (*rho* = 0.7973) (Figure 6), which indicates that the MDS indicators could well represent the TDS. The SQI results showed that soil quality was at a relatively low level in $MM_0$ treatment (0.2–0.4) and a moderate level in $MM_1$, $MM_2$, $IM_0$, and $IM_2$ treatments (0.4–0.6). The highest SQI was found in $IM_1$ treatment (0.6–0.8), indicating good soil quality.

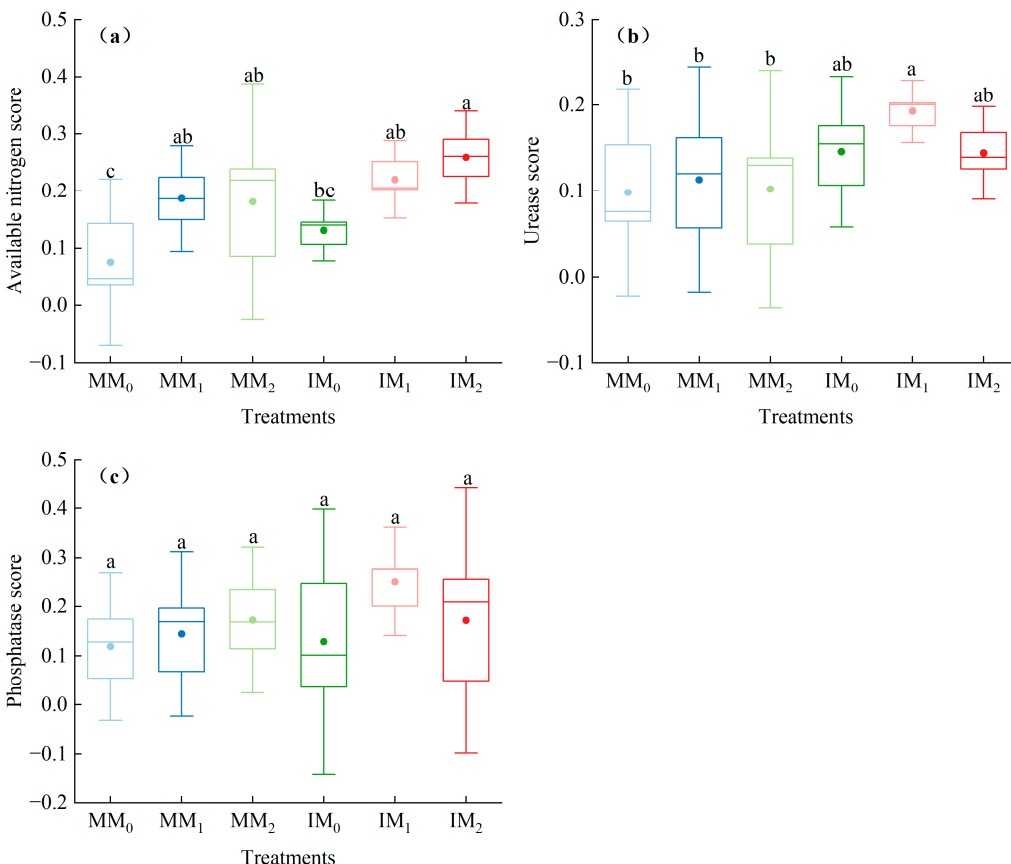

**Figure 4.** Indicator scores of minimum data set for soil quality assessment. (**a**) Available nitrogen score, (**b**) Urease score, and (**c**) phosphatase score. ($MM_0$, $MM_1$, and $MM_2$ represent *L. barbarum* monocropping with zero, medium, and high levels of organic fertilizer, respectively; $IM_0$, $IM_1$, and $IM_2$ represent *L. barbarum/R. sativus* intercropping with zero, medium, and high levels of organic fertilizer, respectively. Error bars represent standard error of the means ($n = 3$). The same lowercase letters above error bars indicate no significant difference between treatments (Duncan's multiple range test: $p < 0.05$).)

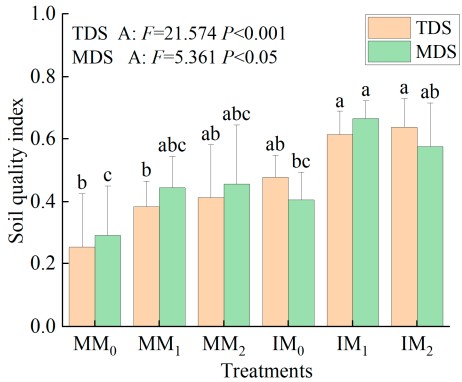

**Figure 5.** Soil quality index of different treatments based on total (TDS) and minimum (MDS) data sets. ($MM_0$, $MM_1$, and $MM_2$ represent *L. barbarum* monocropping with zero, medium, and high levels of organic fertilizer, respectively; $IM_0$, $IM_1$, and $IM_2$ represent *L. barbarum/R. sativus* intercropping with zero, medium, and high levels of organic fertilizer, respectively. Error bars represent standard error of the means ($n = 3$). The same lowercase letters above error bars indicate no significant difference between treatments (Duncan's multiple range test: $p < 0.05$).)

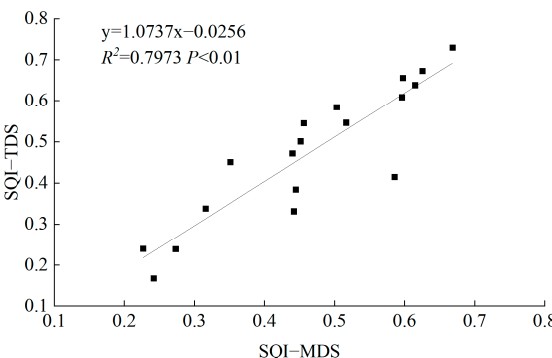

**Figure 6.** Fitting relationship between soil quality index (SQI) values based on minimum (MDS) and total (TDS) data sets.

### 3.5. Soil Quality and Fruit Yield

SEM analysis identified the quantitative relationship between soil management practices and soil quality in terms of individual properties and integrated SQI (*CMIN/DF* = 0.849, *GFI* = 0.881, *CFI* = 1.000, *RMSEA* = 0.000; Figure 7a). Cover cropping had a positive effect on both soil AN (0.46) and TN (0.54), and organic fertilizer mainly contributed to soil AN (0.62). Further, AN (0.54), phosphatase (0.51), and SOC (0.27) had a direct positive influence on SQI.

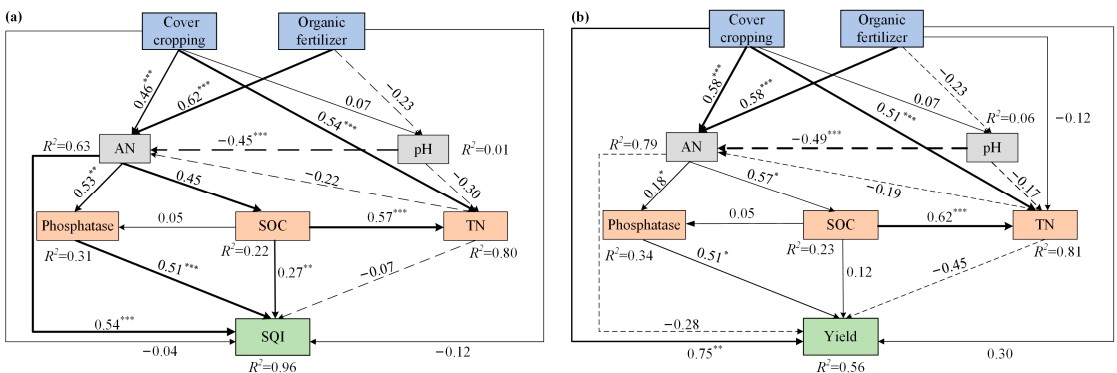

**Figure 7.** Structural equation modeling of factors influencing (**a**) soil quality index (SQI) and (**b**) fruit yield of *Lycium barbarum*. (AN: available nitrogen; SOC: soil organic carbon; TN: total nitrogen. Solid and dashed arrows indicate positive and negative correlations, respectively ($p < 0.05$). Numbers on arrows indicate significant normalized path coefficients (* $p < 0.05$, ** $p < 0.01$, and *** $p < 0.001$).)

A significant effect of cropping system, but not organic fertilizer level, was observed on the fruit yield of *L. barbarum*. The yields from various treatments ranked in the order $IM_1 > IM_2 > IM_0 > MM_2 > MM_0 > MM_1$ (Figure 8). SEM was used to reveal the direct or indirect pathways by which soil management practices drove changes in fruit yield (*CMIN/DF* = 0.582, *GFI* = 0.941, *CFI* = 1.000, *RMSEA* = 0.000; Figure 7b). Cover cropping (0.75) and phosphatase activity (0.51) were significant factors that directly influenced fruit yield. Both cover cropping and organic fertilizer application directly or indirectly affected soil AN, pH, phosphatase, SOC, and TN, thereby influencing fruit yield. In particular, the two soil management practices prominently increased soil AN content (0.58, 0.58), with less direct effects on soil pH (0.07, −0.23). AN, pH, phosphatase, SOC, and TN cumulatively explained 56% of the total yield variation.

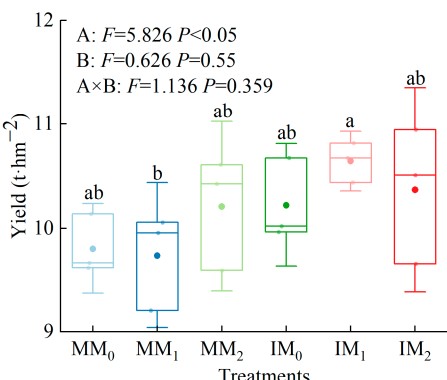

**Figure 8.** Fruit yield of *Lycium barbarum* in various treatments. $MM_0$, $MM_1$, and $MM_2$ represent *L. barbarum* monocropping with zero, medium, and high levels of organic fertilizer, respectively; $IM_0$, $IM_1$, and $IM_2$ represent *L. barbarum/R. sativus* intercropping with zero, medium, and high levels of organic fertilizer, respectively. Values followed by the same lowercase letters in a row are not significantly different between treatments (Duncan's multiple range test: $p < 0.05$).

## 4. Discussion

### 4.1. Amelioration of Soil Physicochemical Properties by Cover Cropping with Organic Fertilizer

During long-term monocropping, soil compaction restricts root elongation, hinders plant growth, and, as such, reduces crop yield [18]. Although soil tillage can temporarily alleviate soil compaction by physically breaking the plow pan, it may destroy the soil structure and result in a decrease in soil fertility [19]. Cover crops, especially deep-rooted species such as *R. sativus*, exhibit a 'bio-tillage' effect because their roots can penetrate the deep layers to alleviate soil compaction [20]. As a consequence of this 'bio-drilling' process, soil BD was distinctly reduced in the *L. barbarum/R. sativus* intercropping system. Moreover, large soil pores areformed after decomposition of *R. sativus*, thereby facilitating water infiltration and air circulation in the soil, which is beneficial to the root growth of the main crops [21].

Cover cropping and organic fertilizer application have great potential to increase soil organic matter content, ameliorate soil fertility, and improve soil health [22]. We found that both soil management practices tended to increase SOC content in the 0–20 cm soil layer of *L. barbarum* orchards, albeit not significantly. The greatest increase in SOC content was observed when *R. sativus* was intercropped with *L. barbarum* under a moderate level of organic fertilizer. Forage radish cover crops can reduce soil nitrogen leaching, as demonstrated by a previous study [23]. Here, cover cropping prominently increased soil TN and AN contents, possibly because *R. sativus* plants captured most of the nitrogen in the soil profile through well-developed deep roots after *L. barbarum* plants stopped absorbing soil nutrients in summer. The roots of *R. sativus* continued to capture the residual nitrogen in soil before dying in winter, and after death, the decomposition of cover crops led to the release of accumulated nitrogen in spring. Therefore, cover cropping not only provides a nitrogen source for subsequent crop growth [24], but it also prevents excess nitrogen leaching into groundwater that may cause environmental pollution [25]. Furthermore, organic fertilizer application increased soil AN content in *L. barbarum* orchards, as it could enhance the soil nitrogen supply by increasing the mineralizable nitrogen pool [26].

After 6 years of *L. barbarum* planting, cover cropping and organic fertilizer application maintained strong acid and alkali buffering capacities in the soil, as no noticeable changes occurred in soil pH between various treatments [27]. Regardless of cover cropping, soil EC increased as a result of increasing level of organic fertilizer. Since a high EC value has a deleterious effect on crop health, farmers should reduce the excessive input of organic fertilizer in *L. barbarum* orchards to maintain a high level of soil fertility and eliminate the constraint of soil salinity. *R. sativus* roots can secrete abundant organic acids that play a role in dissolving phosphorus and potassium [28]. Both cover cropping and organic

fertilizer application increased soil AP and AK contents in *L. barbarum* orchards, which may be attributed to the organic matter decomposition and release of nutrients as well as the transport of nutrients from the deep subsoil to the soil surface by the growth and subsequent decomposition of the cover crop.

*4.2. Enhancement of Soil Biochemical Properties by Cover Cropping with Organic Fertilizer*

Cover cropping and organic fertilizer application had significant positive effects on soil MBC and MBN in *L. barbarum* orchards, which corroborates previous findings in farmland [29,30]. First, the inputs of *R. sativus* plant litter and root exudation could support soil nutrient cycling and provide a substrate that consequently increases microbial structural and functional diversity [31]. Second, cover cropping might enhance soil aggregation and thereby improve the hydraulic properties, water infiltration, and moisture retention, which in turn increases soil heat and air fluxes [32]. These created a favorable soil microenvironment for the growth and reproduction of microorganisms [33], thereby increasing soil microbial activity. In addition to sufficient nutrients, soil microorganisms also need energy supplies—carbon sources. Because organic fertilizer, such as sheep manure used in this study, contains abundant organic matter, it would supplement the carbon and energy sources required for maintenance of microbial growth, ultimately increasing the total soil microbial biomass [34].

Soil enzyme activity is a sensitive indicator of soil nutrient cycling and microbial metabolic activity. In *L. barbarum* orchards, cover cropping considerably enhanced soil enzyme activities, which depends on a range of factors such as fertilizer type and application level, soil type, crop species, soil management, and farming patterns, as well as soil moisture and other environmental conditions [35]. The positive effect of cover cropping on soil sucrase activity is similar to previous results from a maize–green manure intercropping system [36]. Because *R. sativus* features extensive root distribution, cover cropping created favorable environmental and material conditions for microbial activity and consequently facilitated enzyme secretion [37]. Regardless of cover cropping, soil urease activity was highest under a moderate level of organic fertilizer, most likely due to the limited soil carrying capacity. Excessive organic fertilizer application might partially inhibit soil biochemical reactions, as demonstrated in rape fields [38].

*4.3. Improvement of Soil Quality by Cover Cropping with Organic Fertilizer*

Sustainable management and utilization of land resources necessitates accurate soil quality assessment [39]. While no unified assessment standard or method is available worldwide, several studies have used the MDS method to select soil quality indicators [10]. By selecting representative MDS indicators, only a small number of variables need to be determined, which saves manpower and material resources [17]. We selected soil AN content and urease and phosphatase activities as MDS indicators, thus compatible with the selection in a previous study [10]. Soil AN is directly related to plant growth and nutrient supply. Soil urease and phosphatase activities reflect the cyclic transformation of nitrogen and the limitation of soil phosphorus, respectively. Our selection of MDS indicators has implications for holistic assessment of soil quality in characteristic economic crops in the arid areas of Ningxia.

Integrating multiple soil variables into an SQI allows us to evaluate the effect of cultivation practices on soil fertility [9] and then optimize agricultural management. Based on the SQI results ($IM_1 > IM_2 > MM_2 > MM_1 > IM_0 > MM_0$), cover cropping with reduced organic fertilizer improved soil quality (mainly AN and TN) more effectively than organic fertilizer application alone (mainly AN). The combination of cover cropping with a moderate level of organic fertilizer achieved the greatest effect on improving soil quality, which was consistent with previous findings in plantations converted from subtropical forest [13]. While reducing the level of organic fertilizer, cover cropping is beneficial to sustainable soil management in *L. barbarum* orchards.

### 4.4. Yield Validation of Soil Quality Assessment

Crop yields are affected by geographic location, cover type, and staple crop species, soil conditions, and management patterns [23]. In the Argentine Pampas, maize yield was reduced by 8% under non-legume cover cropping and increased by 7% under legume cover cropping compared with fallow controls [40]. In Wisconsin, USA, cover cropping resulted in a 6–9% reduction in maize yield [41]. In our study, cover cropping with a moderate level of organic fertilizer substantially increased *L. barbarum* yield. The yield improvement is attributable to the spatiotemporal complementarity effect of *L. barbarum* and *R. sativus* in terms of peak water and fertilizer demands. Given their minimal interspecific competition, these two crops are suitable for interplanting. Additionally, the roots of *R. sativus* have an outstanding ability to penetrate the soil, known as 'bio-drilling'. Chen and Weil [21] have reported that the well-developed fleshy roots of *R. sativus* can pass through compact soil, leaving a deep root channel. This allows the fibrous roots of the follow-up crop—maize—to easily enter the deep soil and absorb more water, thus increasing maize yield. Furthermore, Lawley et al. (2011) [42] showed that planting *R. sativus* would not reduce the yield of subsequent crops compared with rye or the fallow control.

To validate the reliability of the soil quality assessment results based on MDS, we analyzed the fruit yield of *L. barbarum* in different SQI intervals. We found that cover cropping exhibited a significant positive effect on fruit yield. With an increasing SQI value, fruit yield increased in most treatments, except $MF_1$. Additionally, the positive correlation between fruit yield and SQI confirmed that the soil quality assessment results based on MDS were reliable. SEM revealed that cover cropping, but not organic fertilizer application, directly and significantly increased *L. barbarum* yield, with soil AN, pH, phosphatase, SOC, and TN being the key influencing factors. Since farming habits and climatic conditions all affect *L. barbarum* yield, these factors need to be taken into account in future work.

## 5. Conclusions

This study selected soil urease activity, available nitrogen content, and phosphatase activity as representative indicators to assess soil quality in the *Lycium barbarum* planting system, which effectively reduced the workload. The soil quality index based on a minimum data set was higher under *L. barbarum/Raphanus sativus* intercropping than under traditional monocropping. Cover cropping resulted in higher greatercrop yield, and reduced requirement for organic fertilizer required. Cover cropping with a moderate level of organic fertilizer provided the highest soil quality and crop yield. This soil management pattern provides an effective strategy for ameliorating soil quality and increasing fruit yield in *L. barbarum* orchards in arid areas. The findings of this study provide guidance for soil management in characteristic cash crops in Northwest China. As research is still in its infancy regarding the effects of cover crops on the *L. barbarum* planting system, future work should be carried out from multi-perspectives, such as the duration of cover cropping, soil depth, and environmental factors.

**Author Contributions:** The research work was conceived and designed by F.W., R.R.W. and X.N.; Field work was carried out by X.N., F.W. and W.L.; Soil physicochemical analyses were performed by F.W., W.L. and H.C.; Data was analyzed by X.N., W.L., F.W. and L.Z.; The manuscript was drafted by F.W., W.L., H.C. and revised by X.N. and R.R.W. All authors have read and agreed to the published version of the manuscript.

**Funding:** This research was funded by the National Natural Science Foundation of China (Grant Nos. 42067022; 41761066), the Natural Science Foundation of Ningxia (Grant No. 2022AAC03024), and the Postgraduate Course on the Ideological and Political Dimension (Grant No. KCSZ202203).

**Data Availability Statement:** The datasets supporting the results presented in this manuscript are included within the article.

**Conflicts of Interest:** The authors declare no conflict of interest.

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
