# Peer review of "Forage Radish Cover Crops Improve Soil Quality and Fruit Yield of Lycium barbarum L. in an Arid Area of Northwest China"

_agronomy, doi:10.3390/agronomy13061634_

Round 1
Reviewer 1 Report
The manuscript entitled "Forage radish cover crops improved soil quality and increased fruit yield of Lycium barbarum L. in arid areas of Northwest China" addresses an important thematic, the study of cover crops as an alternative to the conventional practices and their effects on soil quality. Despite the subject of this manuscript is very interesting, it needs to be carefully reformulated and improved in several issues. The Introduction section should be improved, some sentences are very confusing, making it difficult to the reader to understand. The Methods section is well structured and described, however some important data is missing. The Results section needs to be improved, namely the figures size.
- In abstract section the authors present two different nomenclatures for treatments: M0, M1, M2 and F0, F1, F2. The nomenclature F0, F1 and F2 is not present throughout the manuscript. The authors should uniformize nomenclature of treatments.
- Line 46-48. The authors should reformulate this sentence.
- Line 48-50 should be reformulated. Repair that “in orchards” is repeated twice.
- Line 53-56. This sentence is very confusing. Note that “Lycium barbarum L” is repeated several times in the same sentence.
- Line 69. The authors should rearrange this sentence. “Soil quality assessment, to assess…”. Assessment, to assess?
- Line 80. Chen (11). According to citation rules, the authors should cite according to the format: Chen et al (Year) (11). The same for the other citations present in this format.
- Material and Methods section: The description of treatments that are shown in L280-286, should be well explicit in Material and Methods section.
- Important information is missing in Material and Methods section, for example regarding to the yield assessment. How do the authors evaluate the yield? When? In the final of experiment?
- Results section: The figures 2, 3e, 4 and 8, are too small. The authors need to increase the size of figures.
- Figure 7 represents the yield of different treatments. The authors should mention the year when the yield was assessed. As the experiment refers to the assessment of soil quality after six years, it would be important show the cumulative yield, to understand the effects of soil management on yield in each year, instead show only the results of last year (I supposed that refers to the last year).
- Line 490-491: The authors should pay attention to the sentence “planting cover crops helps soil heat preservation”. This can lead to misinterpretations by the reader. Cover crops contributes to moisture retention, contrarily to tillage and other practices which contribute to the increase of soil heat.
- Line 534. The point 4.4 is in the same line that the previous paragraph.
- References section: In some references the Journal format is presented in the abbreviated form, while in other references the Journal name is presented in extense format. The authors should uniformize all the references with the Journal name in the abbreviated format, according to the agronomy instructions for authors.
- Please revise the language of this manuscript, once presents several grammar mistakes. An extensive editing of English language is required.
Reviewer 2 Report
Please open the attached file

the English Language is accepted
Round 2
Reviewer 1 Report
It is notable that the authors made an effort to increase the quality of manuscript. The manuscript can be accepted in the present form.
Reviewer 2 Report
The authors made all the corrections and replied well to all questions. Therefore, I recommended publishing the following manuscript in the Agronomy-Basel journal.
English Language is accepted